# Enzymatic Hydrolysis of Resorcylic Acid Lactones by an *Aeromicrobium* sp.

**DOI:** 10.3390/toxins16090404

**Published:** 2024-09-19

**Authors:** Shawn J. Hoogstra, Kyle N. Hendricks, David R. McMullin, Justin B. Renaud, Juhi Bora, Mark W. Sumarah, Christopher P. Garnham

**Affiliations:** 1London Research and Development Center, Agriculture and Agri-Food Canada, London, ON N5V 4T3, Canada; shawn.hoogstra@agr.gc.ca (S.J.H.); justin.renaud@agr.gc.ca (J.B.R.);; 2Department of Biology, Western University, London, ON N6A 3K7, Canada; 3Department of Chemistry, Carleton University, Ottawa, ON K1S 5B6, Canada; davidmcmullin@cunet.carleton.ca

**Keywords:** zearalenone, HZEN, radicicol, heat shock protein 90 (Hsp90), *Aeromicrobium* sp., α/β-hydrolase, biotransformation, resorcylic acid lactone

## Abstract

Zearalenone and radicicol are resorcylic acid lactones produced by numerous plant pathogenic fungi. Zearalenone is a non-steroidal estrogen mimic that can cause serious reproductive issues in livestock that consume contaminated feed. Radicicol is a potent inhibitor of the molecular chaperone Hsp90, which, in plants, has an important role in coordinating the host’s immune response during infection. Here, we describe the identification and characterization of a soil-borne strain of the Gram-positive bacterium *Aeromicrobium* sp. capable of hydrolyzing the macrolide ring of resorcylic acid lactones, including zearalenone and radicicol. Proteomic analysis of biochemically enriched fractions from the isolated and cultured bacterium identified an α/β-hydrolase responsible for this activity. A recombinantly expressed and purified form of the hydrolase (termed RALH) was active against both zearalenone and radicicol. Interpretation of high-resolution mass spectrometry and NMR data confirmed the structures of the enzymatic products as the previously reported non-toxic metabolite hydrolyzed zearalenone and hydrolyzed radicicol. Hydrolyzed radicicol was demonstrated to no longer inhibit the ATPase activity of the *Saccharomyces cerevisiae* Hsp90 homolog in vitro. Enzymatic degradation of resorcylic acid lactones will enable insight into their biological functions.

## 1. Introduction

Resorcylic acid lactones are structurally related and functionally diverse polyketide secondary metabolites produced by numerous fungi, many of which are plant pathogens. In general, resorcylic acid lactones contain a β-resorcylic acid (typically 2,4 dihydroxybenzoic acid) fused to a macrolactone ring of varying size [1,2]. Resorcylic acid lactones display a wide range of biological activities including antibacterial, antimalarial, antiviral, and anticancer. Many 14-membered resorcylic acid lactones have been reported in the literature, including the hypothemycins [3], agialomcyins [4], and pochonins [5], but the two most important members of this group are zearalenone (ZEN) and radicicol (RAD). 

ZEN (Figure 1) was first discovered in 1962 in extracts from moldy maize [6] and is produced by several species of *Fusarium*, notably *F. graminearum* and *F. culmorum* [7]. ZEN is a common contaminant of maize and wheat and is also detected in other grains including oats and barley [8,9]. This non-steroidal estrogen mimic is capable of binding to the mammalian estrogen receptor [10]. Consumption of ZEN-contaminated feed by livestock can lead to significant reproductive defects, including enlargement and swelling of the ovaries and uterus along with reduced litter sizes and increased stillbirths in swine [11,12]. Management of ZEN from agricultural products is an ongoing problem. ZEN detoxification efforts have included physical, chemical, and biological techniques with varying levels of success [13]. Chemical treatments, including ozonation and oxidation, while effective, can significantly alter the palatability and alimentary qualities of the treated feed [14,15,16]. The same is also true for physical adsorbents that bind and sequester ZEN [16,17]. 

RAD (Figure 1) was first isolated from *Niesslia nordinii* (as *Monocillillium nordinia*) in 1953 [18], and since then has been identified from numerous other fungi, mainly within the *Hypocreales* [19,20]. RAD is a potent and selective inhibitor of Heat shock protein 90 (Hsp90), an ATP-dependent protein-folding chaperone that is highly active during mammalian tumorigenesis [21]. In plants, Hsp90 family members play important roles in stress responses [22]. In particular, during infection, Hsp90 can stabilize plant resistance proteins (R proteins) and assist in the recognition and binding of pathogen effectors, therefore playing an integral role in the host’s innate immune response [22,23]. Several fungi produce RAD and other resorcylic acid lactones as virulence factors when colonizing plants. This includes virulent strains of the fungus *Ilyonectria mors-panacis* during infection of its host crop American ginseng (*Panax quinquefolius*) [24,25,26]. Interestingly, avirulent *Ilyonectria* strains do not produce resorcylic acid lactones. Understanding the biological role of resorcylic acid lactone production is critical as *I. mors-panacis* is responsible for disappearing root rot and replant disease in American ginseng, which poses an existential threat to the ginseng industry [27]. 

Enzymatic degradation of mycotoxins is a recognized mechanism to reduce or eliminate their toxicity [28]. Enzymes that detoxify ZEN via hydrolysis of the lactone moiety are known to produce hydrolyzed ZEN (HZEN) (Figure 1) [29,30]. Both HZEN and its spontaneously decarboxylated metabolite do not bind the estrogen receptor and are non-toxic [31]. A ZEN hydrolase is commercially available as an animal feed additive under the brand name ZEN*zyme*^®^. While ZEN-degrading enzymes are well characterized, to our knowledge, no enzymes that degrade or reduce the activity of RAD have been reported. Based on their structural similarities and enzyme inhibition properties [32], we hypothesize that enzymes that degrade and inactivate ZEN would also be capable of degrading and inactivating RAD and other resorcylic acid lactones. Enzymes that hydrolyze and inactivate RAD could potentially be used to probe the biological function of this family of molecules. Enzymatic degradation of plant pathogenic virulence factors has been demonstrated to reduce the severity of disease. For example, previous research has shown that enzymatic degradation of the virulence factor Triacetyl Fusarinine C (TAFC), a critical iron-acquiring siderophore produced by *I. mors-panacis*, significantly attenuated pathogenic growth of the fungus on ginseng [26]. In this paper, we describe the isolation of a strain of soil-borne Gram-positive bacterium that is capable of degrading ZEN, RAD, and other resorcylic acid lactones from *I. mors-panacis*. Additionally, we characterized the enzyme from this bacterium responsible for hydrolysis activity. 

## 2. Results and Discussion

### 2.1. Identification and Enrichment of ZEN and RAD Hydrolysis Activity

Soil samples were cultured at 28 °C in nutrient broth containing 10 µg/mL ZEN. Following 4 weeks of growth where cultures were refreshed weekly, ZEN degradation was observed in one sample as determined by LC-MS analysis. Individual microbial colonies were isolated on agar plates from that sample and screened for ZEN and RAD degradation, and one colony was identified as being capable of degrading both compounds when grown in liquid culture (Figure 2). When the bacterium was grown in nutrient broth containing ZEN (Figure 2A), a rapid ~3-order-of-magnitude decrease in the ZEN *m*/*z* 317.1394 [M − H]^−^ (C_18_H_22_O_5_) ion peak area was observed after one day. A corresponding increase in peak area for a more polar ion at *m*/*z* 335.1501 (C_18_H_24_O_6_) was also observed that persisted throughout the course of the experiment (Figure 2A). The 18.0107 Da increase in mass (H_2_O) along with an earlier retention time (3.03 min vs. 3.65 min) are consistent with the production of HZEN. Similar results were observed when the isolated bacterium was grown in nutrient broth containing RAD (Figure 2B). A rapid ~1.5 order of magnitude decrease in the RAD *m*/*z* 363.0649 [M − H]^−^ (C_18_H_17_ClO_6_) peak area was observed after one day, together with an associated large increase in peak area for an ion at *m*/*z* 381.0757 (C_18_H_19_ClO_7_) that persisted throughout the experiment. The observed 18.0108 Da increase in mass (H_2_O) with an earlier retention time (2.96 min vs. 3.23 min for RAD) agrees with the conversion of RAD to hydrolyzed RAD (HRAD) (Figure 1). 

Comparison of the 16S rRNA gene of the sequenced bacterium indicated that it was a strain of Gram-positive *Aeromicrobium*. BLAST searches against the NCBI 16S rRNA database showed the sequence was 98.33% identical to *Aeromicrobium choanae* strain 9H-4 (NCBI accession NR_156062.1), 98.11% identical to *Aeromicrobium flavum* strain TYLN1 (NCBI accession NR_044109.1), and 96.37% identical to *Aeromicrobium ginsengisoli* strain Gsoil_098 (NCBI accession NR_041384.1). We refer to the newly identified strain as *Aeromicrobium* sp. LRDC-1. To identify the enzyme responsible for ZEN degradation, ZEN hydrolysis activity was biochemically enriched via Q-Sepharose anion exchange chromatography following a 75% (*w*:*v*) ammonium sulfate precipitation of the clarified cell lysate (Figure 3A). Fractions spanning the entire elution chromatogram were tested for ZEN hydrolysis activity by LC-MS analysis. Fractions displaying ZEN hydrolysis activity eluted discretely off the column and were pooled (horizontal grey bar, Figure 3A) and further fractionated via gel permeation chromatography (Figure 3B). ZEN hydrolysis activity again eluted discreetly, and proteins within active fractions were identified by proteomics analysis. 

The top enzyme candidate from the proteomics analysis was a putative α/β-hydrolase (originally annotated as MenH2, or 2-Succinyl-6-hydroxy-2,4-cyclohexadiene-1-carboxylate synthase) where 15 peptides corresponding to 59.8% sequence coverage of the entire protein were observed (Appendix A). BLASTp searches indicated that the purified protein was homologous to other known ZEN hydrolases. In particular, the putative hydrolase has a 96% identical amino acid sequence to an α/β-hydrolase from a recently discovered *Aeromicrobium* strain capable of degrading ZEN [33] (NCBI accession WP_269305865.1). The next most similar enzyme was another ZEN α/β-hydrolase from *Rhodococcus triatomae* BKS 15-14, which was 77.36% identical (https://patents.google.com/patent/US20220287328A1/en, accessed on 1 July 2024) (NCBI accession EME22619.1). The enzyme was also 63.06% identical to the known ZEN hydrolase from *Rhodococcus erythropolis* (PDB ID 8CLU). We term this newly discovered enzyme resorcylic acid lactone hydrolase (RALH). 

### 2.2. Recombinant RALH Hydrolyzes ZEN and RAD

#### 2.2.1. RALH Hydrolyzes the Lactone Moiety of ZEN

RALH was expressed recombinantly in *E. coli* and purified to homogeneity using a combination of nickel metal affinity, hydrophobic, and gel permeation chromatography steps (Appendix A). Incubation of the purified recombinant RALH with ZEN yielded a dominant [M − H]^−^ ion at *m*/*z* 335.1505 (C_18_H_24_O_6_) with a retention time of 3.01 min (Figure 4A,B). The 18.0122 Da increase in mass of this product, along with its earlier retention time compared to intact ZEN (Figure 4C,D), is consistent with the production of HZEN. It also matches the product observed when ZEN was incubated with the isolated *Aeromicrobium* sp. LRDC-1 bacterium (Figure 2A). Vekiru et al. reported [30] MS/MS product ions for non-toxic HZEN at *m*/*z* 317, 291, 161, and 149, all of which were also observed here (Figure 4B). Interpretation and comparisons of NMR data recorded for ZEN and the enzymatic product confirmed its planar structure as HZEN (Appendix A, Appendix A). Together, these data indicate that RALH from *Aeromicrobium* sp. LRDC-1 can hydrolyze the macrolactone moiety of ZEN to generate HZEN. 

Recently, an *Aeromicrobium* sp. strain capable of degrading ZEN was identified in agricultural soils from the Jiangsu province, China [33]. The hydrolase, ZenH, responsible for this degradation was cloned, expressed, and purified. The authors reported that it degraded ZEN into the putative metabolite, (*S*,*E*)-4-hydroxy-2-(10-hydroxy-6-oxoundec-1-en-1-yl)-7-oxabicyclo [4.2.0]octa-1,3,5-trien-8-one, and not HZEN as we demonstrated here (Figure 4). An amino acid alignment between the hydrolase identified in this study, RALH, and ZenH reported by Hu et al. [33] revealed that their sequences are 96% identical (303/316 identical amino acids) (Appendix A). Homology modelling of the structure [34] indicates that residues that constitute the catalytic triad of the hydrolases—Ser117, His292, and Asp142—are identical, and that all thirteen substitutions between the two enzymes exist within regions not critical for substrate binding or catalysis (Appendix A). Given the similarity of these *Aeromicrobium* sp. and their identified hydrolases capable of inactivating ZEN, it is highly unlikely that, when incubated with ZEN, the two hydrolases are generating different products. Further examination of the MS/MS data for the putative enzymatic product reported by Hu et al. [33] shows that it has the wrong reported molecular formula. This error is likely due to their misidentification of the reported [M + H]^+^ ion, which is actually a [M + H-H_2_O]^+^ ion resulting from the in-source loss of water. Upon examination of the NMR data Hu et al. reported for their putative product, we find it to be nearly identical to our HZEN NMR results (Appendix A, Appendix A) and those reported by Vekiru et al. [30]. Based on this weight of evidence, we would argue that their *Aeromicrobium* sp. hydrolase is also converting ZEN to HZEN.

#### 2.2.2. RALH Hydrolyzes the Lactone Moiety of RAD

Incubation of the purified recombinant RALH with RAD yielded a dominant [M − H]^−^ ion at *m*/*z* 381.0723 (C_18_H_17_ClO_6_) with a retention time of 2.96 min (Figure 5A). Similar to the production of HZEN, the earlier shift in retention time and 18.0102 Da increase in mass compared to RAD (Figure 5C) is consistent with the production of HRAD. These LC-HRMS data are also in accordance with RAD that was incubated with *Aeromicrobium* sp. LRDC-1 (Figure 2B). The MS/MS spectrum of the RALH-catalyzed RAD product showed a prominent neutral loss of 43.989 Da (*m*/*z* 345.0959→301.1063) that corresponds to a loss of CO_2_ (Figure 5B). This strongly suggests that the additional H_2_O occurred by ester hydrolysis of the macrolactone ring as this would lead to the formation of a carboxylic acid moiety (Figure 5B). There is no CO_2_ neutral loss observed in the MS/MS spectrum of RAD (Figure 5D).

#### 2.2.3. NMR Structural Characterization of RAD Degradation Products

To confirm the putative hydrolysis of the RAD lactone moiety by RALH, the major product was purified from the enzymatic mixture and characterized by NMR to confirm its planar structure. NMR spectra recorded for RAD, purchased as an analytical standard (Cayman Chemicals, Ann Arbor, MI, USA), were in accordance with the literature [35] and enabled direct comparisons to the enzymatic product (Table 1) (Appendix A). The enzymatic product was isolated by SPE as a clear solid, had UV absorption maxima at 210 and 270 nm, and the molecular formular C_18_H_17_ClO_6_ determined by an [M − H]^−^ ion at *m*/*z* 381.0748. Interpretation of the ^1^H and HSQC spectrum for the enzymatic product confirmed the presence of one methyl (δ 1.23; H-1), two inequivalent methylenes (δ 1.75/1.62; H-3 and δ 4.27/3.82, H-11), three oxygenated methines (δ 3.95; H-2, δ 3.72; H-5 and δ 3.05; H-4), four olefinic methines (δ 7.69; H-8, δ 6.46; H-7, δ 6.33; H-9 and δ 5.48; H-6), and one aromatic methine (δ 6.35; H-15). Akin to RAD, the ^13^C and HSQC spectra for the enzymatic product revealed that seven of the eighteen carbon signals were not protonated. These seven signals were attributed to a ketone at δ 200.3, a carboxylic acid at δ 175.1 and five sp^2^ carbons at δ 163.8, 156.8, 137.3, 114.5, and 113.2. Unlike RAD, the ^1^H and ^13^C signals for position-11 had low intensity. COSY cross-peaks were observed sequentially from H-1 to H-9 (Figure 5) and key HMBC correlations were observed from H-1 to C-2 and C-3, H-8 and H-9 to C-10, and H-15 to C-13, C-14, and C-16 (Figure 6). These data indicate that the aromatic ring, epoxides, and ketone moieties remain intact for the enzymatic product. The ^1^H and ^13^C NMR data for RAD and the enzymatic product are generally similar except for stark differences around the lactone moiety in RAD (Table 1). For example, the oxygenated methine ^1^H and ^13^C signals for position 2 are shifted from δ 5.38/72.1 in RAD to δ 3.95/66.4 in the enzymatic product. The signals for the lone methyl in RAD at δ 1.52/18.7 are shifted to δ 1.23/23.5 for the RALH product. Similarly, signals for the inequivalent methylene at position 3 are shifted from δ 2.42/1.71/37.7 in RAD to δ 1.75/1.65/42.0. Further, the lactone signal in RAD appears at δ 168.9 (C-18) and is replaced with a characteristic carboxylic acid signal at δ 175.1 (C-18) in the RALH major product. Together, the HRMS and NMR data indicate that RALH hydrolyzes the macrocyclic lactone moiety of RAD to generate HRAD as the major product (Figure 6). Interpretation of the NMR data also indicated that minor products were generated where the epoxides were hydrolyzed. However, we cannot be certain if this is RALH-catalyzed or the result of spontaneous reactions.

### 2.3. HRAD Does Not Inhibit Hsp90

Hydrolytic inactivation leading to detoxification of ZEN is well documented [31]. However, whether hydrolysis of RAD prevents its ability to inhibit its molecular target Hsp90 is unknown. In order to test the activity of HRAD, we measured the ATPase activity of a recombinant form of the *S. cerevisiae* Hsp90 homolog (*Sc*Hsp82) in the presence of both RAD and HRAD using a malachite green assay [32,36]. RAD demonstrated robust inhibition of *Sc*Hsp82, with a 68% decrease in ATPase activity at a concentration of 50 µM (Figure 7). In contrast, no significant inhibition of *Sc*Hsp82 ATPase activity was observed in the presence of 50 µM HRAD. *Sc*HSP82 activity was also significantly attenuated in the presence of 50 µM ZEN, with a 53% decrease in activity, while incubation in the presence of 50 µM HZEN did not significantly inhibit ATPase activity. Taken together, these data indicate that RALH is capable of hydrolyzing the resorcylic acid lactones ZEN and RAD, resulting in significant attenuation of their bioactivity.

### 2.4. RALH Hydrolysis of Resorcylic Acid Lactones from I. mors-panacis Culture Extracts

Previous research has shown that complex mixtures of resorcylic acid lactones, including pochonins and monocillins, are present in culture filtrates of pathogenic *I. mors-panacis* strains [24,35]. While we have demonstrated that RALH can hydrolyze ZEN and RAD, its ability to hydrolyze mixtures of other structurally related resorcylic acid lactones is unknown. We therefore incubated an *I. mors-panacis* culture filtrate extract with RALH and monitored for resorcylic acid lactone hydrolysis by LC-MS analysis (Figure 8 and Appendix A). We observed significant hydrolysis of RAD (98.7% decrease), pochonins D (99.5% decrease) and F (78.2% decrease), and also monocillins II (98.6% decrease), III (98.4% decrease), IV (98.7% decrease), and VI (33.6% decrease).

Biochemical characterization of RALH demonstrated that maximal ZEN hydrolysis activity occurred at pH 8 and a temperature of ~37 °C (Appendix A). Kinetics analysis performed under these conditions determined that RALH is approximately 3.7-fold more efficient at hydrolyzing ZEN compared to RAD. In particular, K_M_ and k_cat_ values of 24 µM and 3.29 s^−1^, respectively, towards ZEN were determined, both of which are in close agreement to previously reported values [33]. K_M_ and k_cat_ values of 1.2 mM and 46.04 s^−1^, respectively, were determined for RAD. These data indicate that RALH binds ZEN with significantly higher affinity but is more efficient at hydrolyzing RAD once bound in the active site. While the core structure of ZEN and RAD are similar, both compounds share obvious structural differences. For RAD, these include the presence of a Cl atom at position 13 and an epoxide moiety bridging positions 4 and 5 (Figure 1). Also, the configuration of the methyl group at position 2 within the macrocyclic ring is different between ZEN and RAD. How all these factors affect the ability of RALH to hydrolyze the two compounds are unknown and requires further structural characterization. While differences in the hydrolysis efficiency of various resorcylic acid lactones are apparent when RALH was incubated with an *Ilyonectria* culture filtrate (Figure 8), the absolute concentrations of each compound in solution was unknown and therefore quantifying relative activities towards each is not possible at this point. Further biochemical work with isolated pochonins and monocillins will also be necessary to understand the substrate specificity of RALH.

Our data demonstrate that HRAD is no longer capable of inhibiting Hsp90 ATPase activity in vitro (Figure 7). It is likely that the hydrolysis of RAD prevents it from interacting at the ATP-binding site, which is the compound’s natural mode of inhibition. These data are not surprising since previous studies have shown that glycosylation of RAD also limits its ability to inhibit Hsp90 [32], likely via steric occlusion. Previous studies have demonstrated that resorcylic acid lactones, including RAD, are produced by pathogenic strains of the fungus *I. mors-panacis*, which causes root rot and replant disease of ginseng [24,35]. The biological function of RAD and other resorcylic acid lactones produced by *I. mors-panacis* is unknown. The ability to target these compounds for enzymatic degradation should provide mechanistic insight into their role during host colonization.

## 3. Conclusions

Here, we describe the identification and characterization of RALH, an α/β-hydrolase characterized from the Gram-positive bacterium *Aeromicrobium* sp. that is capable of hydrolyzing numerous resorcylic acid lactones including ZEN and RAD. NMR and LC-MS analysis demonstrate that RALH clearly hydrolyzes the lactone moiety of both ZEN and RAD. Hydrolysis of RAD prevents it from inhibiting *Sc*Hsp82, the yeast homolog of the protein folding chaperone Hsp90. This information will enable a better understanding of the biological functions of resorcylic acid lactones produced by pathogenic fungi.

## 4. Materials and Methods

### 4.1. Microbial Enrichment and Isolation

Soil samples were collected from several agricultural fields located in SW Ontario and stored at −20 °C in sealed plastic bags until use. Soil samples (0.5 g) were mixed with 3 mL of sterile nutrient broth (NB) (BD Difco, Mississauga, ON, Canada) containing either 300 µM RAD (Cayman Chemical, Ann Arbor, MI, USA) or ZEN (Triple-bond, Guelph, ON, Canada). Cultures were grown aerobically for one week at 28 °C with shaking at 200 rpm. After seven days, each culture was refreshed at a 1:30 ratio into fresh NB media for a total of four weeks. Following this, 100 µL of each culture was mixed with 900 µL methanol and filtered through a 0.45 µm PTFE syringe filter, and resorcylic acid lactone degradation was monitored via liquid chromatography high-resolution mass spectrometry (LC-HRMS) (detailed below). Mixed soil cultures that displayed ZEN and RAD degradation were serially diluted and plated onto NB agar plates. Single colonies were selected and inoculated into fresh NB media and incubated for one week, and RAD and ZEN degradation were once again monitored by LC-HRMS. Cultures displaying ZEN and RAD degradation were transferred three times to ensure a single isolate was obtained. Degradation of ZEN and RAD at 10 µg/mL by the isolated microorganism grown in NB media was monitored daily via LC-HRMS. Bacterial growth was monitored by measuring the OD_600_ of the culture growing in NB media.

### 4.2. LC-HRMS and LC-HRMS/MS Analysis of RAD and ZEN Hydrolysis Products

ZEN and RAD were analyzed using an Agilent 1290 HPLC system coupled to a Thermo Q-Exactive Orbitrap mass spectrometer. All samples were analyzed in negative electrospray ionization mode, with chromatographic separation being performed using an EclipsePlus RRHD C-18 column (2.1 × 50 mm, 1.8 µm; Agilent, Mississauga, ON, Canada). The mobile phase consisted of water with 0.1% formic acid (mobile phase A) and acetonitrile with 0.1% formic acid (mobile phase B) (Optima grade, Fisher Scientific™, Lawn, NJ, USA) at a flow rate of 0.3 mL/min. The gradient was held at 0% B for 30 s, increased to 100% B over 3 min, held at 100% B for 2.5 min, decreased to 0% B over 30 s, and finally held at 0% B for one minute. The injection volume was 5 µL and the column temperature was maintained at 35 °C. The spray voltage was set to 3.5 kV and the probe heater was held at 450 °C. The capillary column temperature was 400 °C, while the nebulizing and auxiliary gases were held at 17 and 8 arbitrary units, respectively. Samples were screened for degradation using a top-5 data-dependent acquisition method, which consisted of a full scan between mass range of 100–900 *m*/*z*, resolution of 35,000, automatic gain control (AGC) of 3 × 10^6^, and a maximum injection time (maxIT) of 128 ms. This was followed by MS/MS scans using a 1.2 *m*/*z* isolation window, 22 normalized collision energy (NCE), 17,500 resolution, 1 × 10^5^ AGC target, and maxIT of 50 ms. All data were analyzed using Thermo Xcalibur software v4.3.

### 4.3. Bacterial Identification and Sequencing

A colony of the isolated bacterium was grown aerobically on NB agar and sent to SeqCenter (Pittsburgh, PA, USA) for de novo genomic sequencing using a combination of Illumina short reads and Oxford Nanopore Technologies (ONT) long reads. Quality control and adapter trimming was performed with bcl-convert v4.3.6 and porechop v0.2.4 (https://emea.support.illumina.com/sequencing/sequencing_software/bcl-convert.html) (https://github.com/rrwick/Porechop) for Illumina and ONT sequencing, respectively. Hybrid assembly with Illumina and ONT reads was performed with Unicycler [37]. Assembly statistics were recorded with QUAST [38]. Assembly annotation was performed with Prokka [39]. The fully sequenced and assembled genome consisted of 3 contigs that were 2,285,130 bp, 885,450 bp, and 208,228 bp in length, for a total genome length of 3,378,808 bp. The GC content of the genome was 71.31%. The nucleotide sequence of the 16S-rRNA gene was blasted against the NCBI 16S ribosomal RNA sequence (bacterial and archaeal) database to identify the microorganism.

### 4.4. Biochemical Enrichment of ZEN Hydrolyzing Activity

The isolated bacterium was inoculated into 20 mL of NB media and incubated at 37 °C for 48 h. The entire culture was then inoculated into 1 L of NB media and grown for an additional 48 h. The culture was centrifuged and the pellet resuspended in 50 mM HEPES (pH 7.0), 50 mM NaCl, 0.2 mg/mL lysozyme, 1.43 mM β-mercaptoethanol, and 2 mM EDTA. The re-suspended bacterial pellet was sonicated to lyse the cells with a cycle of 30 s on and 30 s off for a total of 2.5 min. The lysate was centrifuged (21,000× *g*), and the supernatant was subjected to a 75% (*w*:*v*) ammonium sulfate precipitation. The ensuing protein pellet was resuspended in 50 mM HEPES (pH 7.0) and dialyzed exhaustively against the same buffer at 4 °C. ZEN hydrolase activity of the protein pellet was confirmed by LC-MS screening prior to loading the entire protein pellet on a HiTrap Q HP Column (Cytiva) equilibrated in 50 mM HEPES (pH 7.0) and 50 mM NaCl. Protein was eluted over 10 column volumes using a 50–1000 mM NaCl gradient in 50 mM HEPES (pH 7.0). Active ZEN hydrolyzing fractions were pooled and separated via gel permeation chromatography on an Enrich SEC 650 column (BIO-RAD). The column was equilibrated in 50 mM HEPES (pH 7.0) + 150 mM NaCl. Select fractions displaying ZEN hydrolyzing activity were dialyzed against 50 mM ammonium bicarbonate (pH 8.0) prior to proteomic analysis. All tests for ZEN hydrolyzing activity were conducted at 37 °C overnight with 1 mM ZEN as the substrate.

### 4.5. Proteomics to Identify Candidate Resorcylic Acid Lactone Hydrolases

The proteins within the isolated fractions were identified using mass spectrometry. Aliquots (200 µL) of protein fractions were reduced with 5 mM dithiothreitol (DTT) and incubated at 60 °C for 30 min. The reduced proteins were alkylated at room temperature in darkness using iodoacetamide at a concentration of 10 mM for 15 min. A second addition of DTT was added for a final concentration of 10 mM. The protein fractions were then digested by incubating with 200 ng of trypsin (Thermo Scientific Pierce sequencing grade, Rockford, IL, USA) overnight at 34 °C. The digestion was halted by adding formic acid to a final concentration of 1% (*v*:*v*). The tryptic peptides were purified using a 1 mL Waters Oasis™ hydrophilic–lipophilic balance (HLB) solid phase extraction (SPE) cartridges containing 30 mg sorbent (Milford, MA, USA). The cartridge was first activated with methanol twice and preconditioned twice with LC-MS-grade H_2_O containing 0.1% formic acid. The peptide samples were slowly loaded onto the cartridges, which were dried under vacuum for five minutes. The tryptic peptides were eluted into fresh 1.5 mL microcentrifuge tubes by two 400 μL additions of 70% aqueous acetonitrile. The samples were subsequently dried by vacuum centrifugation (Labconco, Kansas City, MO, USA), reconstituted in 200 μL of water/acetonitrile/formic acid (95/4.9/0.1), and transferred to 250 µL polypropylene HPLC vials. The protein fractions were analyzed using an Easy-nLC™ 1000 nanoflow HPLC system fitted with a 2 cm Acclaim™ C18 PepMap™ trap column and a 75 µm × 15 cm Acclaim™ C18 PepMap™ analytical column (Thermo Scientific) coupled to an Orbitrap mass spectrometer as previously described [35].

Proteomic data across the isolated fractions following gel permeation enrichment were analyzed using MaxQuant v2.1.3.0 [40]. LFQ intensity for each protein was normalized using the highest LFQ intensity across the fractions analyzed. A Pearson correlation between the normalized LFQ intensity to the degradation activity was analyzed for each fraction. The top hits were identified and used to BLAST search for enzymes with putative hydrolase activity.

### 4.6. Recombinant Hydrolase Expression and Purification

The gene encoding the top-ranked ZEN hydrolase candidate (RALH) following proteomics analysis was synthesized and codon-optimized for expression in *E. coli* by Twist Biosciences (www.twistbioscience.com, San Francisco, CA, USA). It was then inserted into the pET His6 MBP TEV LIC cloning vector (Addgene plasmid #29656, a gift from Scott Gradia), placing a Tobacco Etch Virus (TEV) protease cleavage sequence between an N-terminally 6x His-tagged maltose-binding protein (MBP) and the hydrolase. The plasmid was transformed into *E. coli* BL21 (DE3) and grown at 37 °C until the OD_600_ reached 0.5. The temperature was reduced to 16 °C, and protein expression was induced with 750 μM isopropyl β-D-1-thiogalactopyranoside and allowed to proceed overnight at 16 °C with shaking at 200 rpm. Cells were harvested via centrifugation, resuspended in 50 mM Tris HCl (pH 8.0), 500 mM NaCl, and 10 mM β-mercaptoethanol, and lysed via sonication. The lysate supernatant was clarified via centrifugation (21,000× *g*) and subjected to nickel nitrilotriacetic acid metal affinity chromatography. The sample was loaded and washed in 50 mM Tris HCl (pH 8.0), 500 mM NaCl, and 5 mM imidazole (Buffer N) and batch-eluted using Buffer N + 300 mM imidazole. Fractions containing the hydrolase, as determined via SDS-PAGE analysis, were pooled and subjected to TEV protease digestion to remove the MBP fusion tag. Following TEV digestion, (NH_4_)_2_SO_4_ was added at a final concentration of 1M to the sample, which was then loaded onto a Phenyl-Sepharose HP column (Cytiva) equilibrated in 50 mM Tris-HCl (pH 8) and 1M (NH_4_)_2_SO_4_. Protein was eluted using a linear decreasing (NH_4_)_2_SO_4_ gradient (1 to 0 M) in 50 mM Tris-HCl (pH 8) over 10 column volumes. Fractions containing the hydrolase were pooled and subjected to gel permeation chromatography using a HiLoad 16/600 Superdex 200 prep grade column (Cytiva) equilibrated in 50 mM Tris-HCl (pH 8.0) + 150 mM NaCl. Fractions containing the hydrolase were pooled and concentrated to 3.5 mg/mL, flash frozen, and stored at -80 °C until further use. Protein concentration was measured at 280 nm using the enzyme’s mass extinction coefficient (Expasy).

### 4.7. Incubation of RALH with ZEN and RAD

Recombinant RALH (25 nM) was incubated with either 60 µM ZEN or RAD at 25 °C for 45 min in 50 mM Tris HCl (pH 8), 150 mM NaCl, and 1% bovine serum albumin (as carrier protein), and reactions were analyzed via LC-HRMS. Reaction conditions for the production of HZEN and HRAD for NMR structural characterization were as follows: 50 mM Tris HCl (pH 8), 1% bovine serum albumin, 0.4 µg/µL ZEN or RAD, and 20 µM RALH. The reaction was incubated for 72 h at 37 °C. Solid phase extraction (SPE) was used to isolate ZEN and RAD reaction products following incubation with the enzyme using an Oasis HLB 6 cc Vac Cartridge. Following a 1:1 dilution with water, reaction mixtures were passed through the cartridge, washed with 5 mL of water, then eluted stepwise via addition of 60% (*v*:*v*) MeOH:H_2_O and 100% MeOH. the After confirming the presence of HZEN and HRAD, the samples were dried under a gentle stream of N_2_, desiccated for 48 h and weighed prior to NMR analysis.

RALH was tested for ZEN hydrolysis at pH values ranging from 2.2 to 11. Buffers tested were citric acid (pH 2.2 and 3), Sodium acetate trihydrate (pH 4.0), Sodium citrate tribasic dihydrate (pH 5.0), Sodium cacodylate trihydrate (pH 6.0), HEPES sodium (pH 7.0), Tris-HCl (pH 8.0), 3-(Cyclohexylamino)-2-hydroxy-1-propanesulfonic acid (pH 9.0), and N-cyclohexyl-3-aminopropanesulfonic acid (pH 10.0 and 11.0) (Hampton Research, CA, USA). The reaction occurred at 23 °C, with 50 mM buffer, 25 nM RALH, and 60 μM ZEN for 20 min. The effects of temperature on ZEN hydrolysis by RALH were also tested. Reaction temperatures ranged from 23 °C to 80 °C. Reaction volumes were 30 µL, included 25 nM RALH, 60 µM ZEN, and 1% BSA, and occurred for 20 min. Following the reaction, 200 µL 100% MeOH was added to stop the reactions, and each reaction was spun in a centrifuge at 4500× *g* for 45 min. Following this, 50 µL was mixed with 175 µL of 50% MeOH, with 1.92 µM RAD acting as an internal standard (ISTD). RAD has a similar molecular weight and retention time to ZEN, making it a strong ISTD candidate. Activity was then monitored via LC-HRMS.

Kinetics analysis of RALH was performed using the Opentrons OT-2 robot to assist with pipetting H_2_O, 50 mM Tris HCl (pH 8.0), 1% BSA, and ZEN or RAD ranging in concentration from 1 µM to 1500 µM into a 96-well PCR plate. Following this, 25 nM of RALH was added to the plate and incubated at 37 °C for 20 min and 5 min for ZEN and RAD, respectively. Reactions were stopped via the addition of 200 µL of 100% MeOH, and they were then centrifuged at 4 °C and 21,000× *g* for 45 min. Following centrifugation, 30 µL of the quenched reaction was mixed with 270 µL of 50% MeOH. The production of the hydrolyzed product was then monitored via LC-HRMS. The concentration of HZEN or HRAD was quantified using a calibration curve, and the Michaelis–Menten model was applied to the respective curve to determine K_M_ and k_cat_ values.

### 4.8. NMR Structural Characterization of HZEN and HRAD

NMR spectra of native and hydrolyzed resorcylic acid lactones were recorded with a 400 MHz JEOL ECZS Spectrometer (Peabody, MA, USA) with an auto-tuning probe. Resorcylic acid lactones and their hydrolyzed products were dissolved in approximately 500 μL of CD_3_OD (CDN Isotopes, Point Claire, QC, Canada) and referenced to the appropriate solvent peak (δH 3.31, δC 49.1). Comparisons to the literature and interpretation of homonuclear (^1^H, ^13^C, COSY, and NOESY) and heteronuclear (HSQC and HMBC) NMR experiments were used to confirm the structures of the hydrolyzed compounds. ^1^H and ^13^C NMR spectra are reported in the Appendix A.

### 4.9. Expression and Purification of Recombinant Saccharomyces cerevisiae Hsp82 (ScHsp82)

To test the inhibitory activity of RALH-hydrolyzed resorcylic acid lactones, the gene encoding the *Saccharomyces cerevisiae* homolog of Hsp90 (*Sc*HSP82) was synthesized and codon-optimized for expression in *E. coli* by Twist Biosciences (www.twistbioscience.com, CA, USA) and then inserted into the *Nde*I/*Xho*I restriction sites of the pET-28a (+) expression vector, giving the protein an N-terminal 6x-His tag. Protein expression was performed in *E. coli* BL21 cells as described above, and the protein was enriched from the clarified cellular lysate via Ni-NTA metal affinity chromatography as described above. Fractions containing *Sc*HSP82, as determined via SDS-PAGE analysis, were combined and subjected to gel permeation chromatography using a HiLoad 16/600 Superdex 200 prep grade column (Cytiva) equilibrated in 25 mM Tris HCl (pH 7.5) and 50 mM NaCl. Fractions containing the enzyme, as determined via SDS-PAGE, were combined and concentrated to 0.6 mg/mL, flash frozen, and stored at -80 °C until further use. The ATPase activity of *Sc*HSP82 in the presence of various resorcylic acid lactones was determined as previously described [32,36]. Experiments were carried out in quadruplicate at 37 °C in a 96-well plate, with the final reaction conditions as follows: 1 mM ATP, 0.2 µg/µL *Sc*HSP82, 50 µM inhibitor (RAD, HRAD, ZEN, or HZEN), 4% (*v*:*v*) DMSO, 100 mM Tris HCl (pH 7.5), 20 mM KCl, and 6 mM MgCl_2_. Prior to the assay, the isolated resorcylic acid lactones were dissolved in 20% DMSO in assay buffer (100 mM Tris HCl (pH 7.5), 20 mM KCl, and 6 mM MgCl_2_). Reactions were allowed to proceed for 3 h, whereupon the concentration of free inorganic phosphate was quantified using the malachite green colorimetric assay (Sigma-Aldrich, Cat. No. MAK307, Mississauga, ON, Canada). To quantify, 75 µL of assay buffer was added to each well, followed by 20 µL of malachite green working reagent. The plate was shaken and incubated at room temperature for 30 min. Following this, 10 µL of 34% (*w*/*v*) sodium citrate was added to each well to stop the reaction, and absorbance was measured at 620 nm. ATPase activity data was normalized, setting the highest concentration of inorganic phosphate to 100%. Following this, a one-way ANOVA was performed, with a post hoc Tukey’s multiple comparison test, F (4, 15) = 29.81, *p* < 0.001.

### 4.10. RALH Incubation with I. mors-panacis Culture Extracts

Culture filtrate extract containing resorcylic acid lactones was isolated from a pathogenic strain of *I. mors-panacis* as previously described [35]. The extract was adjusted to a final concentration of 0.4 µg/µL and incubated with 20 µM RALH in 50 mM Tris HCl (pH 8). The reaction was incubated for 24 h at 37 °C and stopped via the addition of 100% ice cold MeOH. Samples were subsequently analyzed by LC-HRMS. Compounds were ionized either in positive mode 3.9 kV or negative mode 3.5 kV, in MS1 full-scan mode with 140 K resolution. The flow rate was set to 0.3 mL/min and the gradient elution lasted for 3.5 min. It started with 0% acetonitrile for 1 min and was raised to 100% acetonitrile over 3.5 min. Acetonitrile was kept at 100% for 2.5 min before decreasing to 0% acetonitrile over 0.5 min.

## Figures and Tables

**Figure 1 toxins-16-00404-f001:**
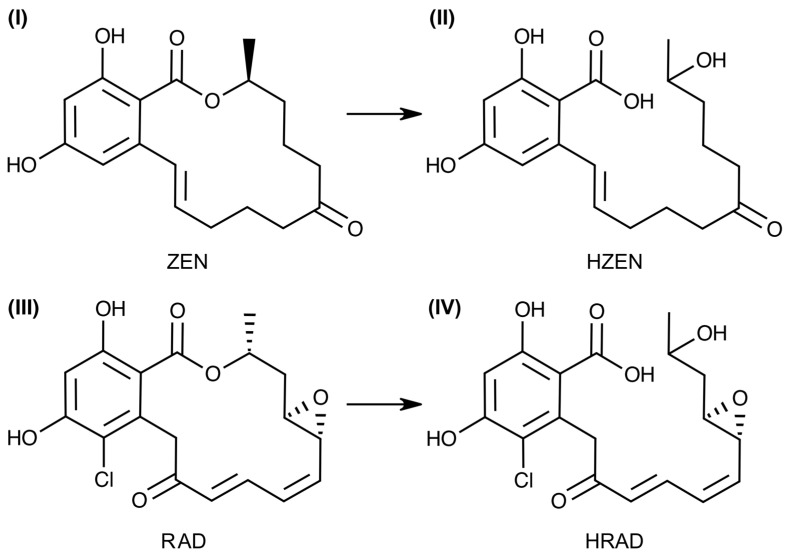
Chemical structures of zearalenone (ZEN) (**I**), hydrolyzed ZEN (HZEN) (**II**), radicicol (RAD) (**III**), and hydrolyzed radicicol (HRAD) (**IV**).

**Figure 2 toxins-16-00404-f002:**
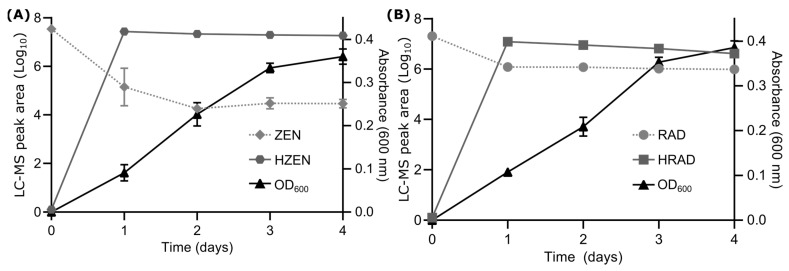
Bacterial degradation of ZEN and RAD. (**A**) ZEN degradation: Solid black line represents the optical density (OD_600_) of the growing bacterial culture. Dashed and gray lines represent LC-MS peak areas (Log_10_) for ZEN (*m*/*z* 317.1394 [M − H]^−^) and HZEN (*m*/*z* 335.1501 [M − H]^−^) ions, respectively. Error bars represent standard deviation (n ≥ 2). (**B**) RAD degradation: Solid black line represents the optical density (OD_600_) of the growing bacterial culture. Dashed and gray lines represent the LC-MS peak areas (Log_10_) for the RAD (*m*/*z* 363.0649 [M − H]^−^) and HRAD (*m*/*z* 381.0757 [M − H]^−^) ions, respectively.

**Figure 3 toxins-16-00404-f003:**
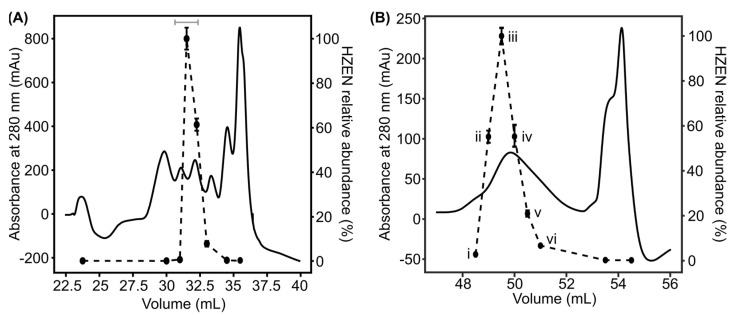
Biochemical enrichment of ZEN hydrolysis activity from *Aeromicrobium* sp. LRDC-1. (**A**) Q-Sepharose ion-exchange enrichment of ZEN hydrolysis activity post ammonium sulfate precipitation. Solid black line represents absorbance at 280 nm, and the dashed line represents the relative abundance of HZEN as determined via LC-MS analysis following incubation of individual fractions with ZEN. The horizontal gray line represents fractions that were pooled for further enrichment. Error bars represent standard deviation (n = 3). (**B**) Gel permeation chromatography enrichment of ZEN hydrolysis activity post Q-Sepharose ion exchange. Black and dashed lines represent the same as panel A. i–vi represent fractions analyzed by LC-MS/MS-based proteomics.

**Figure 4 toxins-16-00404-f004:**
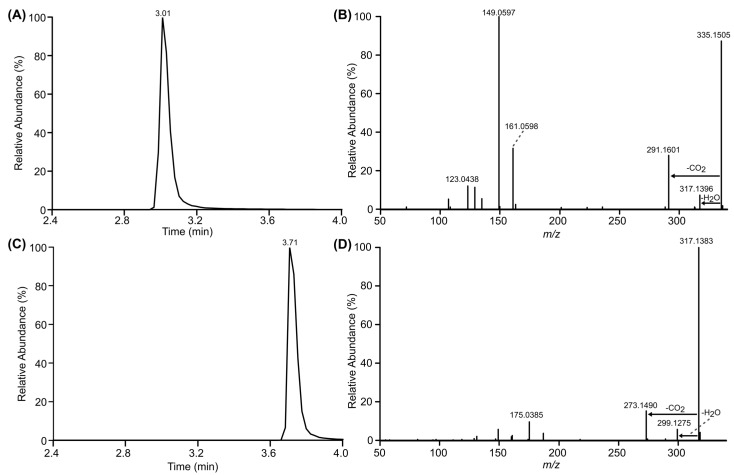
LC-MS and LC-MS/MS analysis of ZEN degradation products following incubation with purified recombinant RALH. (**A**) Extracted ion chromatogram and (**B**) MS/MS spectrum of deprotonated HZEN (*m*/*z* 335.1505 [M − H]^−^). (**C**) Extracted ion chromatogram and (**D**) MS/MS spectrum of deprotonated ZEN (*m*/*z* 317.1383 [M − H]^−^).

**Figure 5 toxins-16-00404-f005:**
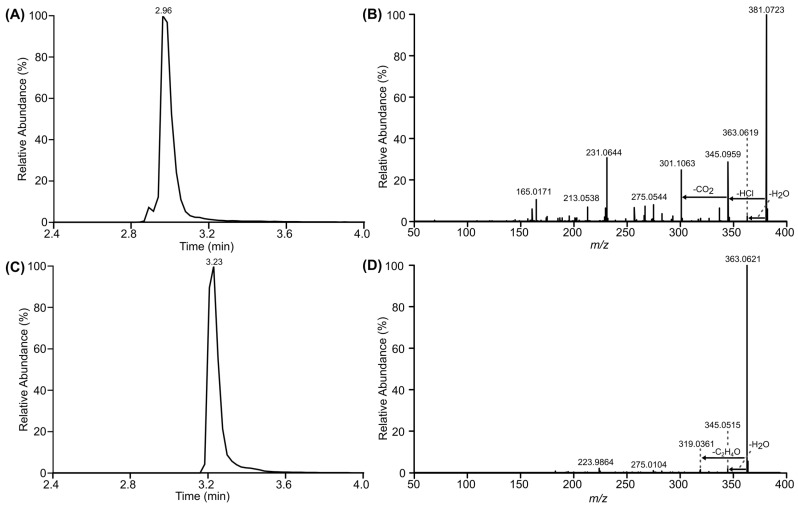
LC-MS and LC-MS/MS analysis of RAD and its hydrolysis product following incubation with purified recombinant RALH. (**A**) Extracted ion chromatogram and (**B**) MS/MS spectrum of deprotonated HRAD (*m*/*z* 381.0723 [M − H]^−^). (**C**) Extracted ion chromatogram and (**D**) MS/MS spectrum of deprotonated RAD (*m*/*z* 363.0621 [M − H]^−^).

**Figure 6 toxins-16-00404-f006:**
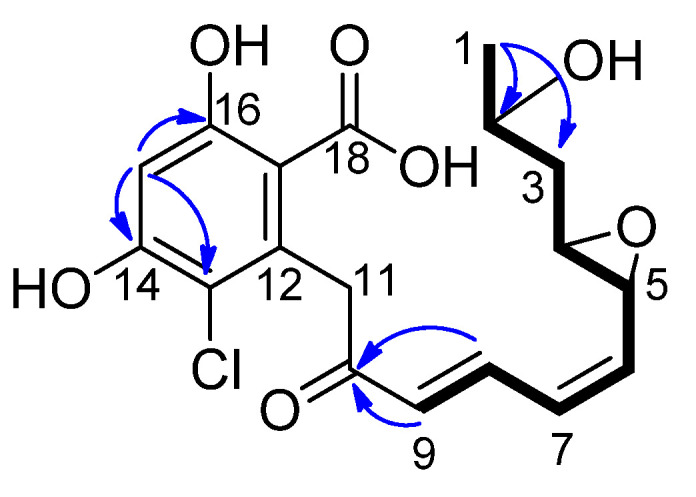
Observed COSY (bold black) and key HMBC (blue arrows) correlations used to decipher the planar structure of HRAD generated by RALH.

**Figure 7 toxins-16-00404-f007:**
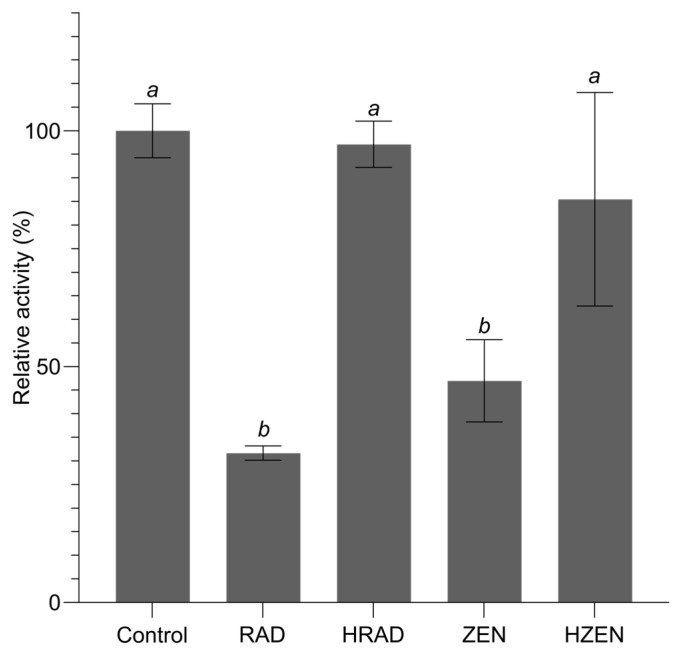
Relative ATPase activity of recombinant ScHsp82 in the presence of 50 µM RAD, HRAD, ZEN, and HZEN. Error bars represent standard deviation (n = 4). Treatments with the same italicized letter are not significantly different by Tukey’s test (*p* < 0.05).

**Figure 8 toxins-16-00404-f008:**
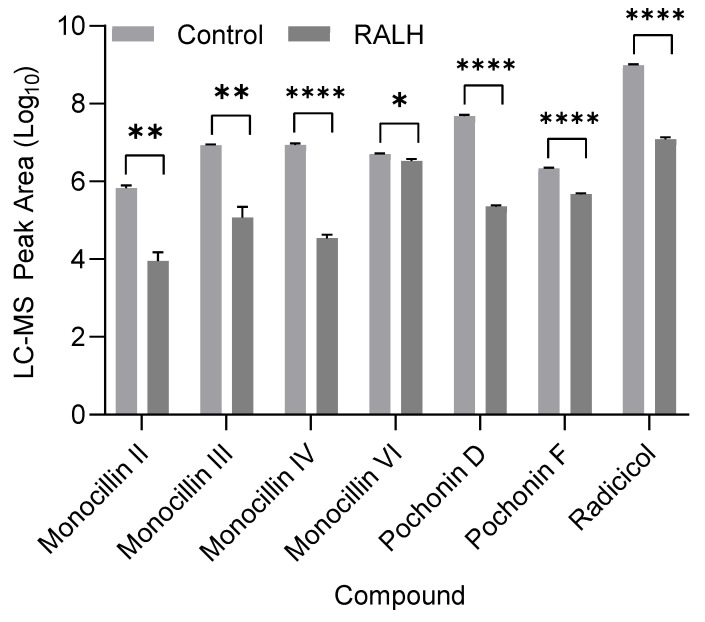
LC-MS peak area (Log_10_) of resorcylic acid lactones from a culture filtrate of *I. mors-panacis* following incubation with RALH. Light grey bars represent negative control (no RALH), while dark grey bars represent extract incubated in the presence of RALH. Error bars represent standard deviation (n = 3). Asterisks represent significance based on two-sample unpaired *t*-test with Welch correction (* = *p* ≤ 0.05; ** = *p* ≤ 0.01; **** = *p* ≤ 0.0001).

**Table 1 toxins-16-00404-t001:** ^1^H (400 MHZ) and ^13^C (100 MHZ) NMR data for RAD and HRAD produced by RALH in CD_3_OD.

	RAD		HRAD	
Position	*δ*_C,_ Type	*δ*_H_ (*J* in Hz)	*δ*_C,_ Type	*δ*_H_ (*J* in Hz)
1	18.7, CH_3_	1.52, d (6.6)	23.5, CH_3_	1.23, d (6.0)
2	72.1, CH	5.38, qt (6.7, 3.7)	66.4, CH	3.95, m
3	37.7, CH_2_	2.42, dt (14.7, 3.5)	42.0, CH_2_	1.75, m
		1.71, ddd (14.7, 8.8, 4.0)		1.62, m
4	56.8, CH	3.06, dt (8.5, 2.8)	59.2, CH	3.05, td (5.7, 2.1)
5	56.5, CH	3.33, o	55.0, CH	3.72, m
6	137.0, CH	5.77, dd (10.8, 4.0)	138.0, CH	5.48, m
7	130.9, CH	6.23, m	130.4, CH	6.46, m
8	140.7. CH	7.59, dd (16.1, 9.8)	138.0, CH	7.69, m
9	131.5, CH	6.09, d (16.1)	132.9, CH	6.33, m
10	199.6, C		200.3, C	
11	46.5, CH_2_	4.15, d (16.3)	48.6, CH_2_ o	4.27, d 15.5)
		3.92, d (16.3)		3.82, d (15.5)
12	136.1, C		137.3, C	
13	115.1, C		114.5, C	
14	159.0, C		163.8, C	
15	103.8, CH	6.47, s	103.4, CH	6.35, s
16	158.0, C		156.8, C	
17	113.5, C		113.2, C	
18	168.9, C		175.1, C	

o—overlap with solvent peak.

## Data Availability

All data are available upon request.

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
