# Peer review of "Enzymatic Hydrolysis of Resorcylic Acid Lactones by an Aeromicrobium sp."

_toxins, 2024, doi:10.3390/toxins16090404_

Round 1

Reviewer 1 Report

Comments and Suggestions for Authors

The authors screened a strain of Aeromicrobium sp. from the soil that can degrade ZEN and RDA. The hydrolase that degrades ZEN and RAD was isolated from the bacterial cell lysate by ammonium sulfate precipitation, anion exchange column enrichment and gel permeation chromatography column separation. The authors analyzed the structures of the degradation products of ZEN and RDA by LC-MS and NMR,. The article has detailed data and clear logic, and is recommended for publication. However, some improvements are still necessary.

Q1: In lines 53-54 of the Introduction section, it is mentioned that RAD inhibits the ATPase activity of Hsp90 and is a potential anticancer drug. This description is not appropriate for this context. In studies on biodegradation, the focus is typically on the harmful effects of the substance being degraded. Including information about its potential benefits could confuse readers. Therefore, it is recommended that the authors remove this description. 

Q2: In lines 289-290, the authors mention the KM and kcat values of the hydrolase RALH, but do not provide information on the enzyme's optimal reaction temperature and pH. These are crucial enzymatic characteristic indicators. Please include these data. 

Q3: In line 282, it is recommended to redraw Figure 8 as a bar graph. 

Q4: Please add the chemical structures of monocillins and pochonins, along with the chemical structures of their metabolites, to the supplementary materials.

Author Response

Comment 1: In lines 53-54 of the Introduction section, it is mentioned that RAD inhibits the ATPase activity of Hsp90 and is a potential anticancer drug. This description is not appropriate for this context. In studies on biodegradation, the focus is typically on the harmful effects of the substance being degraded. Including information about its potential benefits could confuse readers. Therefore, it is recommended that the authors remove this description.

Response 1: We have removed the line from the introduction that states "RAD’s ability to inhibit the ATPase activity of Hsp90 and limit tumorigenesis propelled significant interest in the compound as an anti-cancer drug."

Comment 2: In lines 289-290, the authors mention the KM and kcat values of the hydrolase RALH, but do not provide information on the enzyme's optimal reaction temperature and pH. These are crucial enzymatic characteristic indicators. Please include these data.

Response 2:  We have provided these data in Figure S14 of the supplementary data. We have also updated the text in the main body of the manuscript to describe this data:. It states:

"Biochemical characterization of RALH demonstrated that maximal ZEN hydrolysis activity occurred at pH 8 and a temperature of ~37 oC (Figure S14). Kinetics analysis performed under these conditions determined that RALH is approximately 3.7-fold more efficient at hydrolyzing ZEN compared to RAD." 

We have also added the corresponding text in the methods section to describe how the experiments were performed. It states:

"RALH was tested for ZEN hydrolysis at pH values ranging from 2.2 to 11. Buffers tested were citric acid (pH 2.2 and 3), Sodium acetate trihydrate (pH 4.0), Sodium citrate tribasic dihydrate (pH 5.0), Sodium cacodylate trihydrate (pH 6.0), HEPES sodium (pH 7.0), Tris-HCl (pH 8.0), 3- (Cyclohexylamino)-2-hydroxy-1-propanesulfonic acid (pH 9.0), N-cyclohexyl-3-aminopropanesulfonic acid (pH 10.0 and 11.0) (Hampton Research, CA, USA). The reaction occurred at 23 oC, with 50 mM of buffer, 25 nM of RALH, 60 μM ZEN for 20 min. The effects of temperature on ZEN hydrolysis by RALH were also tested. Reaction temperatures ranged from 23 oC to 80 oC. Reaction volumes were 30 µL and included 25 nM of RALH, 60 µM of ZEN, 1 % BSA and occurred for 20 min. Following the reaction, 200 µL 100% MeOH was added to stop the reactions, and each reaction was spun in a centrifuge at 4,500 × g for 45 min. Following this, 50 µL was mixed with 175 µL of 50% MeOH, with 1.92 µM RAD acting as an internal standard (ISTD). RAD has a similar molecular weight and retention time as ZEN, making it a strong ISTD candidate. Activity was then monitored via LC-HRMS."

Comment 3: In line 282, it is recommended to redraw Figure 8 as a bar graph. 

Response 3: We have updated Figure 8 to a bar graph. 

Comment 4: Please add the chemical structures of monocillins and pochonins, along with the chemical structures of their metabolites, to the supplementary materials.

Response 4: We have provided the structures of the intact and hydrolyzed compounds in the supplementary materials, Figure S15.  

Reviewer 2 Report

Comments and Suggestions for Authors

The authors describe comprehensive research on the specific enzyme from Aeromicrobium sp. for their ability to hydrolyze resorcyclic acid lactones like Zearalenone and Radicicol. Application of various techniques, their detailed description and proper evaluation of the achieved results provide new insights in this area of research. The article is interesting for the respective field of research since it aids in better understanding of the biological functions of resorcyclic acid lactones.

The article is well written and I recommend it for publishing in your Journal.

Author Response

We thank the reviewer for their kind review of the manuscript.